# Virtual Reality in the Neurorehabilitation of Patients with Idiopathic Parkinson’s Disease: Pilot Study

**DOI:** 10.3390/brainsci15101116

**Published:** 2025-10-16

**Authors:** Diana Alejandra Delgado-Anguiano, Ulises Rodríguez-Ortiz, Mireya Chávez-Oliveros, Francisco Paz-Rodríguez

**Affiliations:** 1Universidad Nacional Autónoma de México, Escuela Nacional de Estudios Superiores, Campus León, Guanajuato 37684, Mexico; ale_diana@hotmail.com; 2Instituto Nacional de Neurología y Neurocirugía, Ciudad de México 05120, Mexico; urodriguez@innn.edu.mx (U.R.-O.); mireya.chavez@innn.edu.mx (M.C.-O.)

**Keywords:** Parkinson’s, dance, virtual reality, motor planning, walking speed, falls

## Abstract

**Background:** Parkinson’s disease (PD) is a neurodegenerative condition that affects quality of life due to motor (gait, balance) and cognitive alterations, raising the risk of falling. Virtual reality (RV) and dancing have shown benefits for speed of walking, balance, and postural stability, as well as decreased risk of falls. **Objective**: The goal of this study was to analyze the effectiveness of RV and dancing using a Kinect Xbox 360 video game to improve walking speed and motor performance and reduce the risk of falls in patients with PD. **Method**: This is a pre-experimental study with a simple pre-post design, involving a single group of 14 patients diagnosed with PD in stages 1 to 4 of the Hoehn and Yahr (H and Y) scale, from the National Institute of Neurology (INNN). Before and after the intervention, motor tests, the Unified Parkinson’s Disease Rating Scale (UPDRS-III), the Timed Up and Go (TUG) test, and the Tinetti were applied. The intervention consisted of 16 bi-weekly sessions, which included warm-up, coordination exercises, 10 songs, and cool-down. **Results**: Effects of the RV intervention were observed on improvements in motor tests (z = −2.640, *p* = 0.008), gait (z = −3.316, *p* = 0.001), balance (TUG) (z = −2.966, *p* = 0.001), and on the UPDRS-III scale (total index) (z = −3.048, *p* = 0.002). An increase in the difficulty level of dancing was also observed (X^2^ = 144.13, *p* < 0.01). **Conclusions**: The virtual reality intervention with dancing improved motor performance, including increased walking speed, enhanced postural stability, reduced stiffness and bradykinesia, and a decreased risk of falls

## 1. Introduction

According to the World Health Organization’s definition, Parkinson’s disease (PD) is a neurodegenerative disease with insidious onset [1]. It is diagnosed according to criteria proposed by the UK Brain Bank, based on the presence of four motor symptoms: bradykinesia, resting tremor, rigidity, and postural and gait disturbances [2]. In addition, PD exhibits non-motor symptoms, the most common of which are depression and anxiety, with prevalence rates of 22% and 40%, respectively. Other non-motor symptoms include gastrointestinal problems, such as constipation and decreased sphincter control; sleep disturbances; hyposmia; fatigue; and symptoms related to autonomic dysfunction, such as orthostatic hypotension. The presence of these symptoms reduces PD patients’ independence [3].

PD is caused by the degeneration of nigrostriatal dopaminergic neurons, and motor symptoms become apparent when between 60 and 80% of these neurons in the substantia nigra of the midbrain have been lost, resulting in denervation of the basal ganglia (BG), which affects the motor region (putamen), the associative learning regions (caudate nucleus), and the emotional and reward regions (nucleus accumbens). For all these reasons, PD is not only a motor disorder [4,5].

Gait impairment occurs at all stages of PD and is one of the disease’s most disabling symptoms [6]. Generally, stride length begins decreasing from very early on, and locomotion becomes asymmetrical and begins to slow down. These symptoms tend to be unilateral at first, but become bilateral at moderate stages of the disease, when bradykinesia, freezing of the gait, loss of automaticity, festination, and shuffling can also be observed. These symptoms worsen and become less responsive to pharmacological treatment as PD progresses, which leads to an increased risk of patients falling and eventually leads them to require the assistance of a wheelchair or walker [6,7]. Kearney (2013) reported that the Trail Making Test was the most used measure of executive function, and that poor baseline performance was associated with an increased risk of future falls [8].

Rehabilitation is considered a coadjuvant to pharmacological and surgical treatments that can maximize functional abilities, improve quality of life, and minimize complications. Non-conventional strategies, such as music and dance, as well as interventions that favor social integration, improve postural control, and decrease the risk of falls [9]. Exercise-based video games are a safe, fun, and challenging option for PD patients. These exergames promote body movement and have become one of the best tools for neurorehabilitation [10]. This is because patients can perform fast, long movements involving the entire body. The games require multidirectional movements, weight transfers, a high degree of control and a high number of repetitions. They also require attention, planning, decision-making, sustained concentration, reward, motivation, and commitment to the task performed [11].

Dance in conjunction with virtual reality (VR) has led to positive results in terms of balance, performance of daily activities, and decreasing risk of depression. This is because it provides visual and auditory feedback and increases active learning, motivation, and independence in patients [12]. Dual tasks have been shown to increase walking speed when combined with simple cognitive tasks; therefore, dual motor and cognitive tasks (as seen in VR) provide cognitive benefits in people with PD and healthy adults [13]. Therefore, the aim of this study was to implement a neurorehabilitation program using a virtual reality dance video game (Xbox 360 Kinect) and analyze motor differences before and after the program, as well as identify possible benefits in walking speed, motor performance, and risk of falls in a group of patients with PD.

## 2. Materials and Methods

### 2.1. Sample

An 8-week pre-experimental study was conducted using a simple pre-post design with a single group selected by convenience sampling. The study included 14 patients diagnosed with PD according to the criteria proposed by the UK Parkinson’s Disease Brain Bank and who were classified as being between stages 1 and 4 on the H and Y. None of the patients had been diagnosed with dementia or were suffering alterations to their postural reflexes, and all were able to walk either independently or with help and stand without support for at least 1 min. To be included in the study, patients had to accept certain standards related to regular attendance and active participation and sign informed consent documents. The participants were outpatients treated at the INNN Abnormal Movement Clinic in Mexico City. The study followed the guidelines of the Declaration of Helsinki and was approved by the INNN’s Research and Bioethics Committee. All PD patients were assessed one week before and one week after the neurorehabilitation program by a neurologist (UR), a neuropsychologist (MCH), and a neurological physiotherapist (DD).

### 2.2. Procedure

The treatment was administered in 16 sessions (twice a week). In each session, the patient stood alone in front of a Kinect monitor. The structure of each session was as follows: (1) a warm-up adapted to the patient’s individual capacity; (2) coordination exercises, which were chosen based on the steps that were observed to be the most difficult for them; (3) a dance intervention of 10 songs, with breaks if the patient requested them; (4) stretches, which included the neck (flexion, lateralization, turns), pectorals, latissimus dorsi, hamstrings, and gastrocnemius, with each stretch lasting 30 s.

Video game description: Used in the study were a VR program and an XBOX Kinect 360-based video game called Dance Central 3^®^, developed by Harmonix Music Systems, and released in 2012. The game is played by following a sequence of dance steps, with the user mirroring the on-screen avatar’s sequences. The game has a total of 46 songs, of which 20 from between the 1970s and 2000, were chosen. Songs were selected based on the researchers’ belief that patients would identify them and that this would improve patient adherence. The skill level of each song was established by the video game.

Scores and Stars: The game is based on the correct execution of a sequence of dance steps at a speed that ranges from moderately slow to very fast. The game provides a score and a few stars (maximum 5) at the end of each song. The score is directly related to the number of stars obtained, with a higher score resulting in the participant receiving more stars at the end of the song. The top score available increases as the steps become more precise.

Levels of Ability: The game has four difficulty levels: (1) Beginner, (2) Easy, (3) Medium, and (4) Hard, and each song has a skill level from 1 to 8, with a higher skill level meaning more complex steps. Individual users advanced through the levels of ability when they reached 3 or more stars in 5 or more songs. (This progression was set by the researcher).

Songs: All the songs were danced in “Acting Mode” because this is a single-player mode in which individual participants must mirror the steps of the dance as they are made by the on-screen avatar. Two rounds of 10 songs, each with a duration of 1 min and 30 s, were programmed. In the first round, the skill levels of the songs ranged from 1 to 5, and the songs were arranged in order of increasing difficulty. In the second round, skill levels were between 3 and 7, and again the order of songs was set according to their difficulty.

### 2.3. Instruments and Materials

The Timed Up and Go (TUG) Test: This test is used to measure risk of falls. It is a measure of physical performance that evaluates the ability to stand up from a chair, walk 3 m, turn around, walk towards the chair, and sit down again, measuring the time in which the activity is performed. A participant’s walking speed is assumed normal if the time is ≤10 s, but if they take between 11–20 s or ≥20 s, they are deemed to be at a slight risk of falls or a high risk of falls, respectively. The test has a reliability index of ICC = 0.99 and validity of ICC = 0.99 [14,15].

Tinetti Scale: The Tinetti Scale is used to measure the risk of falls by evaluating the patient’s mobility in two areas: gait and balance. It is composed of nine balance items and seven gait items. Responses are scored 0 if the patient does not achieve or maintain stability in position changes or has an abnormal gait pattern. A rating of 1 means that the patient achieves changes in position and gait patterns but has postural compensations. Finally, a rating of 2 is given to those who do not have any difficulties performing the tasks, and whose response is considered normal. Depending on the total score, the risk of falls is determined: >24 points = *minimum*, 19–24 points = *risk of falls*, and <19 points = *high risk of falls* [16]. In PD patients, the Tinetti test has been found to have a sensitivity of around 76% and a validity of 66% [17].

Unified Parkinson’s Disease Scale (UPDRS-III) motor part: This measurement is used to evaluate the motor symptoms of PD and includes assessment of spoken language, facial expressions, action and rest tremors, rigidity, hand movements, pronation–supination, leg movements, agility, getting up from a chair, posture, gait, postural stability, and bradykinesia. All items have 5 response options: 0 = normal, 1 = mild, 2 = mild-moderate, 3 = moderate, and 4 = severe. The best possible score that indicates that the person is “normal” is 0, and the worst possible score to obtain is 56 points, which indicates severe symptoms [18,19].

In the UPDRS-III motor part [20], motor items are grouped creating subgroups of index:-Axial index: Language, getting up from a chair, gait, posture, retropulsion test (postural stability);-Bradykinesia index: Facial expressions, index finger-thumb tapping, hand opening, pronosupination, lower limb agility, bradykinesia;-Stiffness index: Stiffness (neck, upper limbs (MMSS), lower limbs (MMII);-Tremor index: Rest tremors (MMII, MMSS), action tremors (MMSS).

Montreal Cognitive Assessment (MoCA): This is a cognitive screening test. It is made up of eight areas in which orientation and attention, visuospatial skills/memory, language, abstraction, and executive function are assessed. The maximum score is 30, and the cut-off points for mild cognitive impairment and dementia are 25/26 and 17/18, respectively. The sensitivity of this test in the Mexican population has been shown as 98%, and the specificity as 93% [21].

Patient Health Questionnaire (PHQ-9): This is a screening test to rule out depression. It consists of 9 items that assess the presence and severity of depressive symptoms. According to the scores, the following classification is obtained: (a) major depressive syndrome in those who present 5 or more of the 9 symptoms for more than half the week; (b) negative depressive symptoms in those who do not present the symptoms necessary for diagnosis “more than half the days” in any given period [22].

The two screening tests were used as a filter for the inclusion of patients in the program.

To record scores obtained in the VR program (Scores and Stars), a form organized by song section and session number was created, as well as a register of the current level of ability of each participant. At the end of each session, the stars earned in each song were counted, and if 3 or more stars out of 5 were obtained in 5 or more songs, the level of ability was increased in the following session.

### 2.4. Statistical Analysis

For data analysis, Version 20 of the Statistical Package for the Social Sciences (SPSS Statistics) software suite was used. Central tendency and dispersion measures were obtained for numerical data, as well as frequencies and percentages for categorical variables.

Pretest and posttest data were compared. The distribution of scores was analyzed for each of the outcome variables studied (using the Kolmogorov–Smirnov and Shapiro–Wilk tests), as well as the presence of univariate and multivariate atypical cases (through the analysis of patients’ standardized scores in each of the outcome variables and Mahalanobis distance analysis, respectively).

For differences between demographic variables, the ANOVA test was used. Student’s *t*-test was used for the mean difference in continuous variables. In categorical variables, the Chi-squared test was used; if any had less than a 5% response, Fisher’s exact test was used. To verify compliance with the equality of variances between the pretest and posttest, the Levene test was used.

To evaluate the effect of the experiment, Student’s *t*-test was used for groups related to normal distribution. In cases with abnormal distribution, the non-parametric Wilcoxon Z test was used, including probability values. A 95% confidence interval and Cohen’s effect size (Cohen, J. 1992 [23]) were used. The cutoff points proposed by Cohen were used: values from 0 to 0.2 for small effect size, values close to 0.5 for a medium size, and values close to 0.8 for large effect size [23].

## 3. Results

A total of 30 patients were invited to participate, of which 14 completed the program (Figure 1).

The characteristics of the sample group and its clinical data are shown in Table 1.

### 3.1. Sessions

In relation to the level of dance difficulty reached by the patients, an effect of the intervention was observed (X^2^ = 144.13, gl = 15, *p* < 0.001). Improvement was seen in all 10 songs and followed a similar learning pattern. However, of the 14 participants, only 2 managed to reach the maximum level of ability, and 3 managed to reach Section 2 of songs with high skill levels, taking them to the penultimate level of ability. All patients managed to advance at least one level of ability throughout the 16 sessions (Figure 2).

In skill (score obtained or stars reached in each session), only the song “*In Da Club*” had significant differences (X^2^ = 32.926, df = 15, *p* < 0.005).

### 3.2. Gait

When comparing the walking speed using the Wilcoxon test (−2.003; *p* = 0.001), a significant difference was observed between the initial time of the test (13.14 ± 5.3) and the final time (11.21 ± 2.86), indicating an improvement and increase in speed when standing up from a chair, walking 3 m away, returning, and sitting down again (Table 2).

Regarding the classification of mobility and independence, in the initial evaluation, subjects were categorized into three groups: 2 (14.3%) with independent mobility, 10 (71.4%) mostly independent, and 2 (14.3%) with variable mobility. In the post-intervention assessment, it was observed that five patients from the mostly independent mobility group moved to independent mobility, and two patients who were in the group with more variable mobility moved to the mostly independent group.

Risk of falls, gait, and balance were assessed with the Tinetti scale. The total test score changed from 19.64 ± 4.63 to 26.78 ± 1.89, representing a decreased risk of falls. After treatment, 85.7% (12) of the participants had minimal risk of falling, 14.3% (2) had some risk of falling, and none were at high risk of falling (Figure 3).

Similarly, on the UPDRS-III scale, significant differences were found in the final total score, with 85.7% of patients showing a better score post intervention. Significant differences were also shown in the scores of the indices: axial, bradykinesia, rigidity, and in the total score of the indices. This means that after the intervention, patients presented improvement in the axial index (voice, getting up from a chair, gait, posture, postural stability), bradykinesia (facial expression, the index–thumb test, opening and closing of the hands, pronosupination, leg lifting, and bradykinesia).

## 4. Discussion

Elena et al. (2021), in their systematic review [24], found the use of exergaming as a useful therapy for the general improvement of quality of life in PD, but unlike in the present study, they found no differences between groups in terms gait speed, possibly due to the heterogeneity of the studies and the fact that not all of them evaluated this component.

The present study supports the theory that dance as therapy, in addition to being a safe, accessible, and enjoyable option with good adherence since it does not compromise patient safety, especially in advanced stages of the disease, is beneficial for improving motor performance, mobility, and balance in PD patients [25].

In this study, the video game Dance Central was used because it is based on the precise execution of a sequence of steps, which increase in speed and complexity as the level progresses but does not reach the complexity level of steps performed by professional dancers. As Tillmann (2017) [26] points out, the increase in the speed and complexity of the steps is demanding for users because it requires an increasing ability to execute fast movements. In PD patients, he observed improvements in movement, balance, and cognition when using Samba as an intervention, with varying levels of step speed and increasing complexity [26]. In the present study, patients were able to move up through levels of ability, proving that demanding movement patterns can improve both cognitive and motor functions.

De Natale (2016) demonstrated with an 8-week intervention based on Tango and increased difficulty of steps that patient gait speed in the TUG and a 6-min walk increased more significantly compared to after traditional therapy [27]. An increase in movement speed and a resulting improvement in gait speed were also observed in PD patients in the present study. A decrease in UPDRS-III scores was also observed in this study, which may be due to the multisensory feedback that patients received during the sessions, as well as motor speed and demand, which was related to a decrease in rigidity and bradykinesia.

Van der Kolk (2015) proposed the use of exercise video games in cases of mild–moderate PD, highlighting the importance of the video game as a motivational element when patients receive a reward for the high-intensity training required [28]. In the present case, as well as making greater demands each time, the program gave rewards for advancing through levels (stars), and other motivational stimuli were present among the patients themselves, who motivated the person dancing at the time and congratulated them if they passed a level of ability.

The reduction in limb rigidity (curiously contralateral) may be related to the continuous use of the cross-pattern process, which was found in most of the steps of each song, in addition to constant hand–leg coordination. This result was similar to that reported by Cikajlo et al. (2018) [29].

Also, in the Tinetti test, a decrease in the risk of falls was observed both in the gait and balance subscales and in the total score. Fu et al. (2015) reported the same finding in their study of 30 healthy older adults using the Kinect Wii board as a rehabilitation tool, proving that training programs based on dance and VR have a negative impact on the risk of falls [30].

On the other hand, PD subtypes have significant implications for VR neurorehabilitation feasibility because different subtypes have varying motor, cognitive, and non-motor symptoms, necessitating tailored VR interventions to optimize effectiveness and address specific patient needs. For example, people with PD who predominantly experience tremors may need different VR challenges than those with postural instability. In addition, cognitive deficits in other subtypes could limit their ability to interact with complex VR environments, affecting the feasibility and adaptations necessary for successful treatment and research [31].

Finally, another benefit of this type of therapy is adherence to rehabilitation programs both at six months [32] and one year after the first round of treatment [33]. Although adherence to treatment was not evaluated in the present study, at the end of the intervention, patients requested to continue with the workshop, emphasizing that they had never participated in this type of training before. They reported feelings of recovery and achievement in various skills they had previously given up on, such as performing a turn, a movement many participants indicated they had lost several years ago. Some patients reported that they practiced dancing at home even without the Kinect, helping their motor learning and demonstrating their adherence to the intervention.

## 5. Conclusions

Virtual reality with dance is an effective potential alternative therapy. It is a safe, low-cost, and interactive option that increased walking speed, reduced the severity of motor symptoms such as neck stiffness, limb stiffness, and bradykinesia, improved postural stability in motor performance, and decreased the risk of falls in a group of PD patients. Two 45 min sessions per week for 8 weeks was sufficient to observe benefits. In addition to this, despite initial skepticism on the part of doctors about the ability of participants to advance through the levels of ability and handle the increase in complexity, patients were indeed able to advance and improve, with some of them reaching the highest levels of the game, demonstrating their motor learning and the improvement of their reaction times and coordination.

In conclusion, the treatment applied in this study is effective in managing the symptoms associated with PD, including its various manifestations and subtypes, which tend to become more complicated over time as the effectiveness of pharmacological treatment decreases. This progression has a significant impact on the patient’s quality of life and independence. In the same way, this study is a call to continue the use of innovative new technologies and dance in older patients since it is convenient to find options that encourage patients to carry out their activities independently.

### Limitations of the Study and Suggestions for Further Research

We consider that the most important limitations of this study are the convenience sample size and the absence of a control group that would allow us to attribute the observed effect to the intervention. This can only suggest the presence or absence of an association, although we believe that the results are indeed a consequence of the intervention. We consider that studies with larger samples that include a control group are necessary to determine whether the association is causal in nature. On the other hand, unfortunately, the initial and post-intervention tests were conducted over a very short period (three months). It is recommended that future researchers increase the number of evaluations of the program’s effects to at least three in one year to determine whether its benefits are sustained in the long term.

This study did not encompass the assessment of fear of falling or adherence to treatment, it is recommended that these variables are incorporated into subsequent research. Moreover, it would be prudent to implement a structured program of this nature for patients diagnosed with PD in rehabilitation institutions and clinics. This initiative is expected to enhance motivation levels among this demographic.

Something observed at the end of the research period and that may have been decisive for the achievement of its objectives is that, in the waiting room before starting individual interventions, a group that allowed the socialization and integration of patients and their relatives was formed. This was referred to by patients in an informal way at the end of the intervention, with the majority commenting that this socialization had been important to better performance and increased motivation and improved their perceived quality of life. Therefore, a suggestion is to include the study of psychosocial aspects in interventions like this one in the future.

## Figures and Tables

**Figure 1 brainsci-15-01116-f001:**
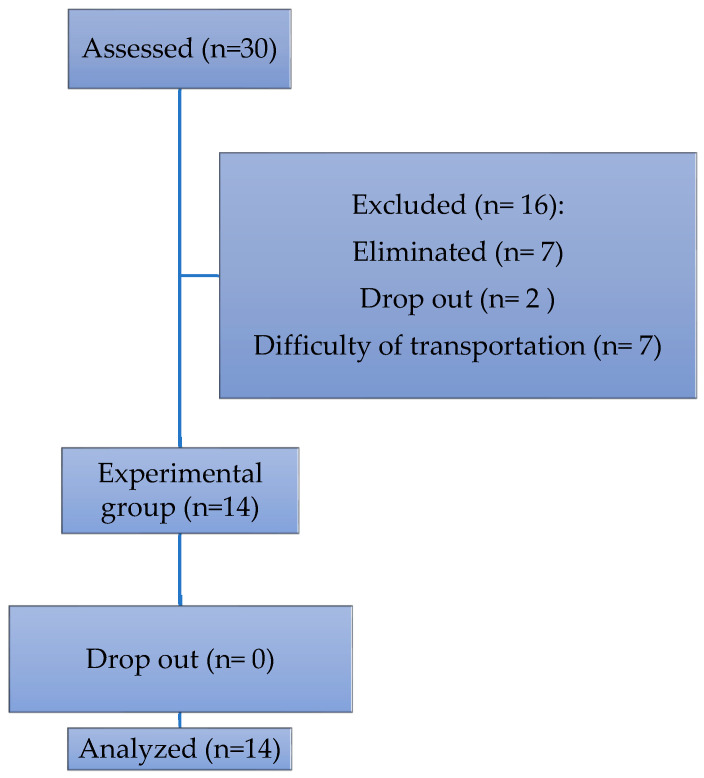
Flowchart for the sample selection.

**Figure 2 brainsci-15-01116-f002:**
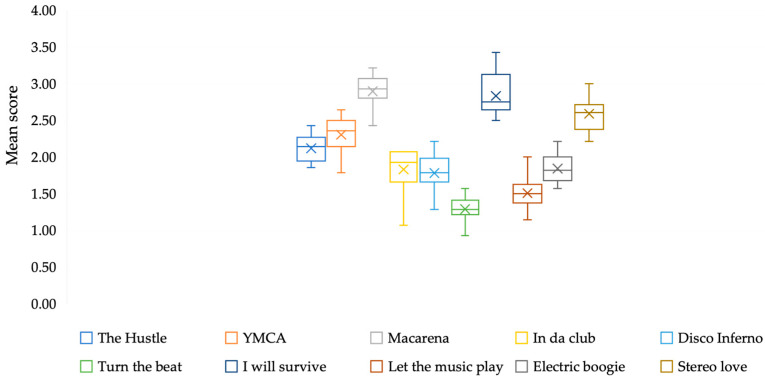
Scores obtained by participants in the dance sessions.

**Figure 3 brainsci-15-01116-f003:**
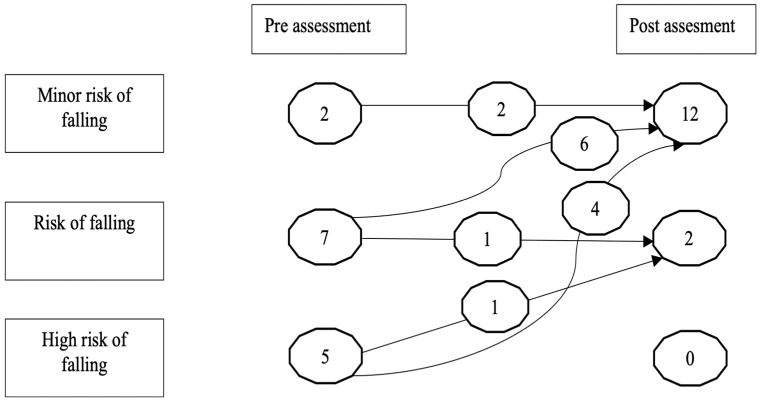
Subject’s risk of falling pre- and post-assessment’s differences from Tinetti’s Test.

**Table 1 brainsci-15-01116-t001:** Demographic and clinical features of sample.

	*n*	%
*Gender*		
Female	9	64.3
Male	5	35.7
*Symptom onset side*		
Left	6	42.9
Right	8	57.1
*H and Y*		
Unilateral symptoms (I)	1	7.1
Bilateral mild symptoms (II)	3	21.4
Mild–moderate bilateral symptoms (III)	10	71.5
	M	SD
*Age*	68.43	8.1
*Years of study*	10.92	4.74
*Disease duration (years)*	9.93	7.71
*Dyskinesias*		
Yes	5	35.7
No	9	64.3
*Dyskinesias duration (years)*	1.79	3.09
*MoCA*	26.31	1.97

M = mean, DS = standard deviation, S = sample, % = percentage.

**Table 2 brainsci-15-01116-t002:** Comparing the walking speed in pre- and post-assessment.

Results	Pre (N = 14)		Post (N = 14)				
	M (SD)	CI (95%)	M (DS)	CI (95%)	Z	Sig.	Cohen d
**Gait**							
TUG	13.14 (5.43)	(10.01, 16.28)	11.21 (2.86)	(9.56, 12.87)	−2.003	0.001	0.53
Tinetti: balance	12.14 (2.98)	(10.42, 13.87)	15.07 (1.44)	(14.24, 15.90)	−2.966	0.001	0.79
Tinetti: gait	7.50 (2.18)	(6.24, 8.76)	11.71 (0.73)	(11.30, 12.13)	−3.316	0.001	0.89
Total Tinetti Test Score	19.64 (4.63)	(16.97, 22.32)	26.78 (1.89)	(25.70, 27.88)	−3.299	0.001	0.88
**UPDRS**							
Total Score	26.00 (8.26)	(21.23, 30.77)	20.00 (7.58)	(15.63, 24.37)	−3.047	0.002	0.82
Rigidity: neck	1.57 (1.02)	(0.99, 2.16)	0.86 (0.66)	(0.47, 1.24)	−2.640	0.008	0.71
Rigidity: right upper limb	1.29 (0.83)	(0.81, 1.76)	0.57 (0.51)	(0.28, 0.87)	−2.640	0.008	0.71
Rigidity: left lower limb	1.29 (0.83)	(0.81, 1.76)	0.64 (0.84)	(0.16, 1.13)	−2.714	0.007	0.76
Pull test	1.14 (0.86)	(0.64, 1.64)	0.79 (0.70)	(0.38, 1.19)	−2.236	0.025	0.60
Axial index score	4.86 (2.60)	(3.36, 6.36)	3.57 (2.38)	(2.20, 4.94)	−1.975	0.048	0.53
Bradykinesia index score	13.29 (3.56)	(11.23, 15.34)	12.14 (3.23)	(10.28, 14.01)	−1.980	0.048	0.53
Rigidity index score	6.29 (3.58)	(4.22, 8.35)	3.29 (2.64)	(1.76, 4.81)	−2.952	0.003	0.79
Total index score	26.07 (8.13)	(21.38, 30.77)	20.14 (7.54)	(15.79, 24.50)	−3.048	0.002	0.82

Note: Pre (pre-assessment), Post (post-assessment), M (mean), CI (confidence interval), Sig. (significance), TUG (Timed Up and Go Test), UPDRS (Unified Parkinson’s Disease Rating Scale).

## Data Availability

The data presented in this study are available on request from the corresponding author. The data are not publicly available due to ethical restrictions.

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
