# Peer review of "Virtual Reality in the Neurorehabilitation of Patients with Idiopathic Parkinson’s Disease: Pilot Study"

_brainsci, 2025, doi:10.3390/brainsci15101116_

Round 1

Reviewer 1 Report

Comments and Suggestions for Authors

Delgado-Anguiano et al elaborate on the significance of virtual reality in the neurorehabilitation of patients with idiopathic Parkinson's disease. I have the following comments:

  • in the introduction authors mention the pathogenesis of parkinsonisms, it would be valuable to indicate the possible vascular factor - Ref. The Significance of Vascular Pathogenesis in the Examination of Corticobasal Syndrome. Frontiers in aging neuroscience13, 668614. https://doi.org/10.3389/fnagi.2021.668614
  • though authors classify this research as a pilot study, the possible significance of the interpretation of the results of 14 patients should be more justified
  • was the levodopa equivalent dose of patients verified
  • authors could elaborate on the possible significance of PD subtypes in the context of feasibility of virtual reality neurorehabilitation
  • figure 3 is difficult to read and should be corrected

Author Response

Virtual reality in the neurorehabilitation of patients with idiopathic Parkinson's disease. Pilot study. (brainsci-3900151)

We thank all the useful comments, we believe this new version of our manuscript meets all of them. Please see the attachment

Revisor 1

Comments 1: in the introduction authors mention the pathogenesis of parkinsonisms, it would be valuable to indicate the possible vascular factor - Ref. The Significance of Vascular Pathogenesis in the Examination of Corticobasal Syndrome. Frontiers in aging neuroscience13, 668614. https://doi.org/10.3389/fnagi.2021.668614

Response 1:  Thank you for your comment. The reference was consulted and added to the text, which now reads: “It is known that PD is caused by the degeneration of nigrostriatal dopaminergic neurons, with a possible vascular origin as the cause [4]”

Comments 2: though authors classify this research as a pilot study, the possible significance of the interpretation of the results of 14 patients should be more justified

Response 2:  Our results from 14 patients showed that dancing through VR, in addition to being a safe, accessible, and viable option, improves motor performance, mobility, and balance. It also provides benefits in terms of mood and quality of life for PD patients.

Comments 3: was the levodopa equivalent dose of patients verified

Response 3: It was verified and there was no equivalence in the doses.

Comments 4: authors could elaborate on the possible significance of PD subtypes in the context of feasibility of virtual reality neurorehabilitation

Response 4: Thank you for your comment. As indicated Fereshtehnejad, PD subtypes have significant implications for VR neurorehabilitation feasibility because different subtypes have varying motor, cognitive, and non-motor symptoms, necessitating tailored VR interventions to optimize effectiveness and address specific patient needs. For example, people with PD who predominantly experience tremors may need different virtual reality (VR) challenges than those with postural instability. In addition, cognitive deficits in other subtypes could limit their ability to interact with complex VR environments, affecting the feasibility and adaptations necessary for successful treatment and research. To review the updated version go to lines 302-309.

Fereshtehnejad, S. M.; Postuma, R. B. Subtipos de la enfermedad de Parkinson: ¿Qué nos dicen sobre la progresión de la enfermedad? Curr Neurol Neurosci Rep 2017; 17: 34. https://doi.org/10.1007/s11910-017-0738-x

Comments 5: figure 3 is difficult to read and should be corrected

Response 5: Figure 3 has been redrawn for better understanding. To review the updated version go to lines 254-255.

Reviewer 2 Report

Comments and Suggestions for Authors

General statement: The authors should carefully edit the manuscript in its entirety to ensure a high degree of clarity of expression and eliminate grammatical errors.

Line 6. The text “Pilot, quantitative, comparative…” is not a complete sentence. Please correct this.

Line 20. Please introduce the acronyms UPDRS-III and TUG here on their first use in the Abstract.

Line 21. Consider replacing with “16 bi-weekly sessions”.

Line 22. Presumably, the authors mean “cool down” rather than “cooling”.

Line 22. The phrasing could be improved here. For example, it could be stated as “RV intervention improved motor tests, including…”

Line 25: It is unclear what is meant by “The difficulty level advance was shown in the dance…”. Please clarify.

Line 34. Replace by “…based on the presence of four motor symptoms:”.

Lines 33-35. Although it is unclear as written, do the authors wish to convey here that the Parkinson’s UK Brain Bank proposed the criteria for the clinical diagnosis of PD? Please clarify this.

Lines 35-37. This could be more simply stated. Consider: “In addition, PD has non-motor symptoms. The most common motor symptoms are….”

Line 37. It is incorrect to state a prevalence as a range (i.e., 25-40%). Please correct this.

Lines 40-41. Consider: “The presence of these symptoms decreases the degree of independence of PD patients.”.

Lines 43. Please use either “between 60 and 80%” or “60-80%”.

Line 44. Replace with “substantia nigra pars compacta”.

Lines 44-45. The loss of substantia nigra dopamine neurons results in dopamine depletion from the striatum. The sentence reads as if the basal ganglia has sectors for motor learning, associative learning, and emotion. This is not true. It also appears as though the authors are claiming that the loss of dopamine terminals in the basal ganglia results in an increase in astrocytes and microglia in the basal ganglia. In addition, I didn’t find any mention of these points in Zesiewicz (2019). There are many excellent reviews on the synaptic organization and function of the basal ganglia (PMID: 10923985, 1695400, 20411769). Please consider consulting them.

Lines 47-48. Please provide a citation for the first sentence in the paragraph regarding gait impairment.

Lines 55-56. Do the authors mean to state that poorer performance on the maze test is associated with future falls? Please clarify.

Lines 57-58. Please consider “Rehabilitation is considered a coadjuvant to pharmacological…”. Delete “In dealing with PD,”.

Lines 59-61. Please consider “Non-conventional strategies, such as music, dance, and interventions that favor social integration, improve postural control and decrease fall risk.”

Lines 63-67. This very long sentence contains too many ideas. Please break it up into 2-3 sentences.

Line 68. Please introduce the term “VR” here on its first use in the text.

Line 69-71. This very long sentence contains too many ideas. Please break it up into 2-3 sentences.

Line 73. Please introduce the term “EP” here on its first use in the text.

Line 71. Dual tasks increase the speed what specifically? How does dual tasks connect to the VR task and the motor performance of PD patients?

General comment: The limitations of the use of convenience sampling on the ability of authors to make inferences should be discussed in the “Limitations” subsection (line 326).

Figure 1 legend. Please provide a more descriptive figure legend. Replace “Analized” with “Analyzed”.

Figure 3. This figure is uninterpretable as presented. Please remake the figure. The numerous overlapping arrows are very distracting.

General statement: Although this is an interesting study, the lack of the inclusion of a suitable control group and the use of convenience sampling place important limitations on the conclusions that can be made.

Comments on the Quality of English Language

The authors should carefully edit the manuscript in its entirety to ensure a high degree of clarity of expression and eliminate grammatical errors.

Author Response

General statement: The authors should carefully edit the manuscript in its entirety to ensure a high degree of clarity of expression and eliminate grammatical errors.

We would like to express our sincere gratitude for the time, effort, and expertise you devoted to reviewing our manuscript. Your thoughtful and constructive feedback has been invaluable in enhancing the clarity, rigor, and overall quality of our work.

Comment 1. Line 6. The text “Pilot, quantitative, comparative…” is not a complete sentence. Please correct this.

Response: The sentence was replace by “8-week pre-experimental study with a simple pre-post design involving a single group”. To review the updated version go to lines 17-18

Comment 2. Line 20. Please introduce the acronyms UPDRS-III and TUG here on their first use in the Abstract.

Response: The acronyms were introduced. To review the updated version go to lines 20 - 21.

Comment 3. Line 21. Consider replacing with “16 bi-weekly sessions”.

Response 3: To review the updated version, go to line 22

Comment 4. Line 22. Presumably, the authors mean “cool down” rather than “cooling”.

Response 4: To review the updated version, go to line 23

Comment 5. Line 22. The phrasing could be improved here. For example, it could be stated as “RV intervention improved motor tests, including…”

Response 5: To review the updated version, go to lines 27-28

Comment  6. Line 25: It is unclear what is meant by “The difficulty level advance was shown in the dance…”. Please clarify.

Response 6: To clarify the phrase, we changed the sentence to ‘level of difficulty of the dance’. To review the updated version, go to line 26

Comment 7. Line 34. Replace by “…based on the presence of four motor symptoms:”

Response 7: To review the updated version, go to line 35

Comment 8. Lines 33-35. Although it is unclear as written, do the authors wish to convey here that the Parkinson’s UK Brain Bank proposed the criteria for the clinical diagnosis of PD? Please clarify this.

Response 8: This paragraph was replace by “The diagnosis is made according to the criteria proposed by the UK Brain Bank, which are based on the presence of four motor symptoms: bradykinesia, resting tremor, rigidity, and postural and gait disturbances”. To review the updated version, go to lines 54-57

Comment 9. Lines 35-37. This could be more simply stated. Consider: “In addition, PD has non-motor symptoms. The most common motor symptoms are….”

Response 9: To review the updated version, go to line 37

Comment 10. Line 37. It is incorrect to state a prevalence as a range (i.e., 25-40%). Please correct this.

Response 10: To review the updated version, go to line 38

Comment 11. Lines 40-41. Consider: “The presence of these symptoms decreases the degree of independence of PD patients.”.

Response 11: To review the updated version, go to lines 41-42

Comment 12. Line 43. Please use either “between 60 and 80%” or “60-80%”.

Response12 : To review the updated version, go to line 45

Comment 13. Line 44. Replace with “substantia nigra pars compacta”.

Response 13: To review the updated version, go to line 45-46

Comment 14. Lines 44-45. The loss of substantia nigra dopamine neurons results in dopamine depletion from the striatum. The sentence reads as if the basal ganglia has sectors for motor learning, associative learning, and emotion. This is not true. It also appears as though the authors are claiming that the loss of dopamine terminals in the basal ganglia results in an increase in astrocytes and microglia in the basal ganglia. In addition, I didn’t find any mention of these points in Zesiewicz (2019). There are many excellent reviews on the synaptic organization and function of the basal ganglia (PMID: 10923985, 1695400, 20411769). Please consider consulting them.

Response 14: The sentence was changed by “The basal ganglia are a grouping of interconnected subcortical nuclei that mitigate and control functions ranging from voluntary movement, cognitive planning, emotions and reward functions, and even cognition and learning” and the reference of Zesiewicz (2019) was replaced by Sonne, J; et al. (2025). To review the updated version, go to line 46-48

Comment 15. Lines 47-48. Please provide a citation for the first sentence in the paragraph regarding gait impairment.

Response 15: The citation for the sentence in the paragraph regarding gait impairment was: Mirelman, A.; Bonato, P.; Camicioli, R.; Ellis, T. D.; Giladi, N.; Hamilton, J. L.; Hass, C. J.; Hausdorff, J. M.; Pelonsin, E.; Almeida Q, J. Gait impairments in Parkinson’s disease. Lancet Neurol. 2019, 18(7), 697–708. https://doi.org/10.1016/S1474-4422(19)30044-4

Comment 16. Lines 55-56. Do the authors mean to state that poorer performance on the maze test is associated with future falls? Please clarify.

Response16: In this sentence, the maze test was replaced by the Trail Making Test.  To review the updated version, go to line 57

Comment 17. Lines 57-58. Please consider “Rehabilitation is considered a coadjuvant to pharmacological…”. Delete “In dealing with PD,”.

Response 17: We have considered your suggestion. To review the updated version, go to line 59

Comment 18. Lines 59-61. Please consider “Non-conventional strategies, such as music, dance, and interventions that favor social integration, improve postural control and decrease fall risk”

Response 18: We have considered your suggestion. To review the updated version, go to lines 61-62

Comment 19. Lines 63-67. This very long sentence contains too many ideas. Please break it up into 2-3 sentences.

Response 19: We break it down into two sentences To review the updated version, go to lines 64-69

Comment 20. Line 68. Please introduce the term “VR” here on its first use in the text.

Response 20: We introduced the term “VR”. To review the updated version, go to line 70

Comment 21. Line 69-71. This very long sentence contains too many ideas. Please break it up into 2-3 sentences.

Response 21: We break it down into two sentences To review the updated version, go to lines 70-73

Comment 22. Line 73. Please introduce the term “EP” here on its first use in the text.

Response 22: The term “EP” was replaced by “PD” To review the updated version, go to line 75

Comment 23. Line 71. Dual tasks increase the speed what specifically? How does dual tasks connect to the VR task and the motor performance of PD patients?

Response 23: The Dual tasks increase the walking speed in patients with PD. The use of virtual reality (VR) through the Wii device involves the participation of various processes, such as motor, visual, and attentional. This characteristic makes VR a dual task, suggesting the need for divided attention and simultaneous interaction of different brain areas. To review the updated version, go to lines 73-75

Comment 24. General comment: The limitations of the use of convenience sampling on the ability of authors to make inferences should be discussed in the “Limitations” subsection (line 326).

Response 24: We agree that one of the disadvantages of convenience sampling is that it can lead to biased results, given that participants are selected based on their availability. We consider this in the “limitations subsection”. To review the updated version, go to line 338

Comment 25: Figure 1 legend. Please provide a more descriptive figure legend. Replace.

Response 25: The legend for Figure 1 has been replaced. To review the updated version, go to line 206. The word “Analized” was changed to “Analyzed”, to review the updated version, go to lines line 204

Comment 26: Figure 3. This figure is uninterpretable as presented. Please remake the figure. The numerous overlapping arrows are very distracting.

Response 26: Figure 3 has been redrawn for better understanding. To review the updated version, go to lines 254-255.

Comment 27: General statement: Although this is an interesting study, the lack of the inclusion of a suitable control group and the use of convenience sampling place important limitations on the conclusions that can be made.

Response 27: We agree that one of our main limitations is the sample size, obtained for convenience, and the absence of a control group, which does not allow us to attribute the observed effect to the intervention. Therefore, we can only suggest the presence or absence of an association, although we believe that the result is, in fact, a consequence of the intervention. We believe that studies with larger samples that include a control group are needed to determine whether the association is causal in nature.

Round 2

Reviewer 1 Report

Comments and Suggestions for Authors

Acceptable as it is.

Author Response

We would like to express our sincere gratitude for the time, effort, and expertise you devoted to reviewing our manuscript. Your thoughtful and constructive feedback has been invaluable in enhancing the clarity, rigor, and overall quality of our work.

Reviewer 2 Report

Comments and Suggestions for Authors

General statement: The revised manuscript contains too many grammatical errors. The authors should carefully edit the manuscript in its entirety to ensure a high degree of clarity of expression and eliminate grammatical errors.

Lines 17-18. Although I appreciate the edit, this is still not a complete sentence. It needs to be revised.

Line 21. Please consider replacing with “Timed Up and Go (TUG) test”.

Line 26. Please replace with “χ2” here and wherever it appears in the manuscript.

Line 27. Replace with “motor performance”.

Line 38. Replace with “…22% and 40%, respectively.”.

Line 43. Replace with “PD is caused by the…”

Line 44. The sentence fragment “…with a possible vascular origin as the cause…” is unclear. Do you mean that a causal agent has been observed in the blood? Please clarify.

Line 46. It would be better to discuss the striatum here rather than the basal ganglia as it would connect with the information provided on line 43 (i.e., degeneration of nigrostriatal dopamine neurons).

Lines 65-66. Replace with “…perform fast, long movements involving the entire body.”. There are a lot of complex tasks that do not involve fast, long movements that involve the entire body.

Line 66. Replace with “The games require multi-directional…”.

Lines 73-74. Replace with “tasks (seen in VR) provide…”.

Line 306. Replace with “…may need different VR challenges…”.

Figure 3. Replace “Pos assessment” with “Post assessment”.

Line 257. Replace with “post assessment, 12 of…”.

Figure 2. In the figure, most of the symbols corresponding to the different songs (e.g., Electric boogie) cannot be seen. It might be better to report the mean score ± standard error for each session. The song list on the right hand side can be deleted. “-4” on the x-axis should be deleted. The y-axis title should be “Mean score”. The horizontal grids lines are distracting and can be eliminated. Please move the graph title “Level reached…” to the figure legend.

Comments on the Quality of English Language

The revised manuscript contains too many grammatical errors. The authors should carefully edit the manuscript in its entirety to ensure a high degree of clarity of expression and eliminate grammatical errors.

Author Response

Virtual reality in the neurorehabilitation of patients with idiopathic Parkinson's disease. Pilot study. (brainsci-3900151)

General statement: The revised manuscript contains too many grammatical errors. The authors should carefully edit the manuscript in its entirety to ensure a high degree of clarity of expression and eliminate grammatical errors.

Response: We would like to express our sincere gratitude for your feedback. Following your suggestions, we have reviewed the manuscript once more and corrected the grammatical errors.

Comment 1: Lines 17-18. Although I appreciate the edit, this is still not a complete sentence. It needs to be revised.

Response 1: A modification has been made to the sentence structure. This is a pre-experimental study with a simple pre-post design, involving a single group of 14 patients diagnosed with PD in stages 1 to 4 of Hoehn and Yahr”

Comment 2: Line 21. Please consider replacing with “Timed Up and Go (TUG) test”.

Response 2: The suggested change was made. Timed Up and Go (TUG) test

Comment 3: Line 26. Please replace with “χ2” here and wherever it appears in the manuscript.

Response 3: The suggested change was made

Comment 4: Line 27. Replace with “motor performance”.

Response 4: The suggested change was replacedThe virtual reality intervention with dancing improved motor performance”

Comment 5: Line 38. Replace with “…22% and 40%, respectively.”.

Response 5: The suggested change was replaced. “with prevalence rates of 22% and 40%, respectively.”

Comment 6: Line 43. Replace with “PD is caused by the…”

Response 6: The suggested change was replaced

Comment 7: Line 44. The sentence fragment “…with a possible vascular origin as the cause…” is unclear. Do you mean that a causal agent has been observed in the blood? Please clarify.

Response 7: Following a thorough review of the paragraph, it was decided to make changes to the way it was written. “PD is caused by the degeneration of nigrostriatal dopaminergic neurons, and motor symptoms begin to become apparent when between 60 and 80% of these neurons in the substantia nigra of the midbrain have been lost, resulting in denervation of the basal ganglia (BG) that affects the motor region (putamen), the associative learning regions (caudate nucleus), and the emotional and reward regions (nucleus accumbens). For all these reasons, PD is not only a motor disorder”.

Comment 8: Line 46. It would be better to discuss the striatum here rather than the basal ganglia as it would connect with the information provided on line 43 (i.e., degeneration of nigrostriatal dopamine neurons).

Response 8: With the change made in the previous paragraph, we consider this question to be resolved.

Comment 9: Lines 65-66. Replace with “…perform fast, long movements involving the entire body.”. There are a lot of complex tasks that do not involve fast, long movements that involve the entire body.

Response 9: The suggested change was replaced

Comment 10: Line 66. Replace with “The games require multi-directional…”.

Response 10: The suggested change was replaced

Comment 11: Lines 73-74. Replace with “tasks (seen in VR) provide…”.

Response 11: The suggested change was replaced

Comment 12: Line 306. Replace with “…may need different VR challenges…”.

Response 12: The suggested change was replaced

Comment 13: Figure 3. Replace “Pos assessment” with “Post assessment”.

Response 13: The suggested change was replaced

Comment 14: Line 257. Replace with “post assessment, 12 of…”.

Response 14: The suggested change was replaced

Comment 15: Figure 2. In the figure, most of the symbols corresponding to the different songs (e.g., Electric boogie) cannot be seen. It might be better to report the mean score ± standard error for each session. The song list on the right hand side can be deleted. “-4” on the x-axis should be deleted. The y-axis title should be “Mean score”. The horizontal grids lines are distracting and can be eliminated. Please move the graph title “Level reached…” to the figure legend.

Response: The suggested change was made.